# HyperDense-Net: A hyper-densely connected CNN for multi-modal image segmentation

**Jose Dolz**[1]                                                    jose.dolz@etsmtl.ca
**Karthik Gopinath**[1]                                    karthik.gopinath.1@etsmtl.net
**Jing Yuan**[2]                                                    jyuan@xidian.edu.cn
**Hervé Lombaert**[1]                                        herve.lombaert@etsmtl.ca
**Christian Desrosiers**[1]                          christian.desrosiers@etsmtl.ca
**Ismail Ben Ayed**[1]                                        ismail.benayed@etsmtl.ca

[1] *ÉTS Montreal, Canada*

[2] *Xidian University, School of Mathematics and Statistics, Xian, China*

## Abstract

In this work we present *HyperDenseNet*, a recently published 3D fully convolutional neural network that extends the definition of dense connectivity to multi-modal segmentation problems (Dolz et al., 2018). In this architecture each modality is processed by a separate path, and dense connections occur not only between the pairs of layers within the same path, but also between those across different paths. Hence, the proposed network has total freedom to learn more complex combinations between the different modalities, *within and in-between all the levels of abstraction*, increasing significantly the learning representation. *HyperDenseNet* is evaluated on two challenging multi-modal brain tissue segmentation datasets, iSEG 2017 and MRBrainS'13, yielding significant improvements over many state-of-the-art segmentation networks on both benchmarks. Our code is publicly available[1].

**Keywords:** Deep learning,brain MRI,segmentation,3D CNN,multi-modal imaging

## 1. Introduction

Recently, convolutional neural networks (CNNs) have been adopted as an effective alternative to traditional segmentation approaches. To tackle with multi-modal imaging, existing CNN segmentation techniques follow either an *early-fusion* (Moeskops et al., 2016; Kamnitsas et al., 2017; Chen et al., 2017a) or a *late-fusion* (Nie et al., 2016) strategy, where low-level or high-level features are fused, respectively. In either case modeling several modalities relies entirely on a single joint layer, or level of abstraction, limiting the representation learning capabilities of deep models in this scenario. On the other hand, densely connected networks (Huang et al., 2017) have become very popular in many vision problems. Such a dense connectivity facilitates the gradient flow and the learning of more complex patterns, yielding significant improvements in accuracy and efficiency for natural image classification tasks (Huang et al., 2017). In the medical domain, recent works have explored the use of dense connections in deep segmentation networks (Yu et al., 2017; Chen et al., 2018). Nevertheless, these works consider either a single modality (Yu et al., 2017) or simply con-

---

1. https://www.github.com/josedolz/HyperDenseNet

catenate multiple modalities in a single stream (Chen et al., 2018). Thus, the impact of dense connectivity across multiple network paths, and its application to multi-modal image segmentation, remains unexplored.

In this abstract we present *HyperDenseNet*, a 3D fully convolutional neural network that extends the definition of dense connectivity to multi-modal segmentation problems. In contrast to previous works, where the fusion stage is defined by the user, we provide the network with total freedom to learn complex combinations of patterns across modalities and at different levels of abstraction. In addition to increase the representation power compared to early/late fusion strategies, hyper-dense connections facilitate the learning as they improve gradient flow and impose implicit deep supervision. This extended abstract is a reduced version of the recently published journal manuscript (Dolz et al., 2018).

## 2. HyperDenseNet

Let $\boldsymbol{x}_l$ be the output of the $l^{th}$ layer. In CNNs, this vector is typically obtained from the output of the previous layer $\boldsymbol{x}_{l-1}$ by a mapping $H_l$ composed of a convolution followed by a non-linear activation function $\boldsymbol{x}_l = H_l(\boldsymbol{x}_{l-1})$. In densely-connected networks all feature outputs are concatenated in a feed-forward manner, $\boldsymbol{x}_l = H_l([\boldsymbol{x}_{l-1}, \boldsymbol{x}_{l-2}, \ldots, \boldsymbol{x}_0])$, where $[\ldots]$ denotes a concatenation operation. Pushing this idea further, *HyperDenseNet* introduces a more general connectivity definition, in which we link the outputs from layers in different streams, each associated with a different image modality. For simplicity, let us consider the scenario of two image modalities, although extension to $N$ modalities is straightforward. Let $\boldsymbol{x}_l^1$ and $\boldsymbol{x}_l^2$ denote the outputs of the $l^{th}$ layer in streams 1 and 2, respectively. In general, the output of the $l^{th}$ layer in a stream $s$ can then be defined as $\boldsymbol{x}_l^s = H_l^s([\boldsymbol{x}_{l-1}^1, \boldsymbol{x}_{l-1}^2, \boldsymbol{x}_{l-2}^1, \boldsymbol{x}_{l-2}^2, \ldots, \boldsymbol{x}_0^1, \boldsymbol{x}_0^2])$. Furthermore, shuffling and interleaving feature map elements in a CNN has been recently found to enhance the efficiency and performance, while serving as a strong regularizer (Zhang et al., 2017; Chen et al., 2017b). We thus concatenate feature maps in a different order for each branch and layer, resulting in $\boldsymbol{x}_l^s = H_l^s(\pi_l^s([\boldsymbol{x}_{l-1}^1, \boldsymbol{x}_{l-1}^2, \boldsymbol{x}_{l-2}^1, \boldsymbol{x}_{l-2}^2, \ldots, \boldsymbol{x}_0^1, \boldsymbol{x}_0^2]))$. A section of the proposed architecture is depicted in Fig. 1.

**Network architecture.** *HyperDenseNet* is inspired by the baseline architecture proposed in (Kamnitsas et al., 2017). As in (Dolz et al., 2017), we employ 9 convolutional layers with 25,25,25,50,50,50,75,75 and 75 kernels of size 3×3×3, and three fully convolutional connected layers with 400, 200 and 150 1×1×1 kernels, respectively. We employ sub-volumes of size 27×27×27 for training and 35×35×35 non-overlapping sub-volumes for inference. For the rest of hyper-parameters we refer to the detailed explanations in (Dolz et al., 2018).

## 3. Experiments and results.

The proposed *HyperDenseNet* architecture is evaluated on two challenging multi-modal image segmentation tasks, using publicly available data from two MICCAI Challenges: infant brain tissue segmentation –iSEG (Wang et al., 2019)– and adult brain tissue segmentation –MRBrainS 2013 (Mendrik et al., 2015)–. These tasks consist on automatically segmenting the brain into three main classes: white matter (WM), grey matter (GM) and cerebro-spinal

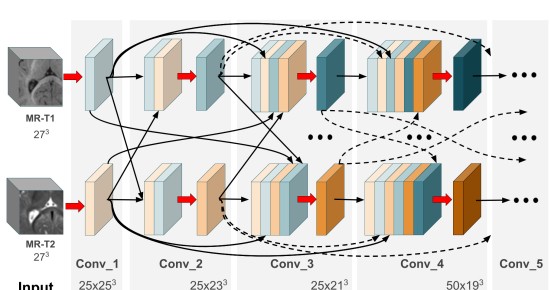

Figure 1: A section of the proposed Hyper-DenseNet with two image modalities.

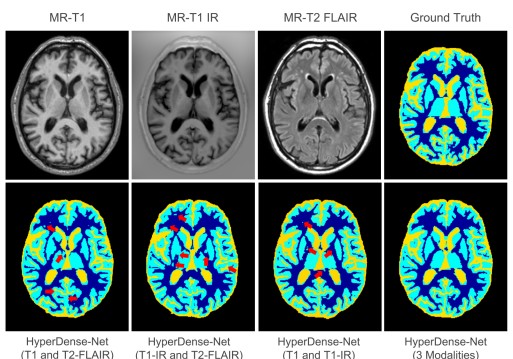

Figure 2: Visual results achieved by HyperDenseNet.

fluid (CSF). Quantitative evaluations and comparisons with state-of-the-art methods are reported for each of these applications. For this purpose we report Dice Similarity Coefficient (DSC), Modified Hausdorff distance (MHD) and Average Surface Distance (ASD) values, for the iSEG segmentation results. On the other hand, for the case of the MRBrainS'13 challenge, we resort to the DSC, MHD and absolute volume difference (AVD) metrics. In both cases, the MHD corresponds to the 95th-percentile of the Hausdorff distance.

Table 1: Results on iSEG-2017 and MRBRains 2013* data for HyperDenseNet and the methods ranked in the top-5.

| Method | CSF | | | GM | | | WM | | |
|---|---|---|---|---|---|---|---|---|---|
| | DSC | MHD | ASD | DSC | MHD | ASD | DSC | MHD | ASD |
| *iSEG 2017* | | | | | | | | | |
| BCH_CRL_IMAGINE | **0.960** | 8.850 | **0.110** | **0.926** | 9.557 | **0.311** | **0.907** | 7.104 | **0.360** |
| MSL_SKKU | 0.958 | 9.112 | 0.116 | 0.923 | 5.999 | 0.321 | 0.904 | **6.618** | 0.375 |
| **HyperDenseNet (Ours)** | 0.956 | 9.421 | 0.120 | 0.920 | **5.752** | 0.329 | 0.901 | 6.660 | 0.382 |
| LIVIA (ensemble) | 0.957 | 9.029 | 0.138 | 0.919 | 6.415 | 0.338 | 0.897 | 6.975 | 0.376 |
| Bern_IPMI | 0.954 | 9.616 | 0.127 | 0.916 | 6.455 | 0.341 | 0.896 | 6.782 | 0.398 |
| | DSC | MHD | AVD | DSC | MHD | AVD | DSC | MHD | AVD |
| *MRBrains 2013** | | | | | | | | | |
| **HyperDenseNet (ours)** | **0.8633** | **1.34** | 6.19 | **0.8946** | **1.78** | 6.03 | 0.8342 | 2.26 | 7.31 |
| VoxResNet (Chen et al., 2017a)+Auto-context | 0.8615 | 1.44 | 6.60 | 0.8946 | 1.93 | 6.05 | **0.8425** | **2.19** | 7.69 |
| VoxResNet (Chen et al., 2017a) | 0.8612 | 1.47 | 6.42 | 0.8939 | 1.93 | 5.84 | 0.8396 | 2.28 | 7.44 |
| MSL-SKKU | 0.8606 | 1.52 | 6.60 | 0.8900 | 2.11 | **5.54** | 0.8376 | 2.32 | **6.77** |
| LRDE | 0.8603 | 1.44 | **6.05** | 0.8929 | 1.86 | 5.83 | 0.8244 | 2.28 | 9.03 |

*The reported values were obtained from the challenge organizers at the time of submitting the original manuscript, in February 2018.

Table 1 reports the segmentation performance of HyperDenseNet compared to that of other top ranking methods of the iSEG and MRBrains 2013 Challenge. We observe that the proposed network ranked among the top-3 methods in 6 out of 9 metrics, considering the results of the iSEG Challenge. Similarly, HyperDenseNet achieved the best results in 4 out of 9 metrics, being on the top-3 in 8 of them in the MRBrainS'13 dataset. We also show visual results of HyperDenseNet in Fig. 2, where segmentations obtained with the combination of different image modalities are shown.

## Acknowledgments

This work is supported by the National Science and Engineering Research Council of Canada (NSERC), discovery grant program, and by the ETS Research Chair on Artificial Intelligence in Medical Imaging.

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
