# OpenReview forum: "HyperDense-Net: A hyper-densely connected CNN for multi-modal image segmentation"
_MIDL.io/2019/Conference/Abstract — MIDL Abstract 2019_

### Official Review · AnonReviewer1 · 2019-04-24
**Extend the densenet to multi-modal**

**Rating:** 3
**Confidence:** 3

**Review:**

The paper is well written with plenty of validations with well-known challenges.
The core idea is to extend the densenet to multi-modal.
The idea is neat and useful when multiple modalities are available.
It would be better to describe the computational resource required for this work.

---

### Official Review · AnonReviewer2 · 2019-04-25
**Good summary of recently published method**

**Rating:** 3
**Confidence:** 3

**Review:**

This abstract is a well-written summary of the authors' recent IEEE-TMI paper on this network architecture for multi-modaml segmentation with crossed connections. The present results on tissue segmentation on two datasets. This poster will in my opinion generate good discussion.

---

### Decision · Program_Chairs · 2019-05-06
**Acceptance Decision**

Accept